# A Honeybee-Inspired Framework for a Smart City Free of Internet Scams

**DOI:** 10.3390/s23094284

**Published:** 2023-04-26

**Authors:** Abdulghani Ali Ahmed, Ali Al-Bayatti, Mubarak Saif, Waheb A. Jabbar, Taha H. Rassem

**Affiliations:** 1School of Computer Science and Informatics, De Montfort University, The Gateway, Leicester LE1 9BH, UK; 2Faculty of Computer Science and Information Technology, Universiti Tun Hussein Onn Malaysia, Parit Raja 86400, Malaysia; 3School of Engineering and the Built Environment, Birmingham City University, Birmingham B4 7XG, UK

**Keywords:** internet scams, phishing websites, web spoofing, honeybee, smart city

## Abstract

Internet scams are fraudulent attempts aim to lure computer users to reveal their credentials or redirect their connections to spoofed webpages rather than the actual ones. Users’ confidential information, such as usernames, passwords, and financial account numbers, is the main target of these fraudulent attempts. Internet scammers often use phishing attacks, which have no boundaries, since they could exceed hijacking conventional cyber ecosystems to hack intelligent systems, which emerged recently for the use within smart cities. This paper therefore develops a real-time framework inspired by the honeybee defense mechanism in nature for filtering phishing website attacks in smart cities. In particular, the proposed framework filters phishing websites through three main phases of investigation: PhishTank-Match (PM), Undesirable-Absent (UA), and Desirable-Present (DP) investigation phases. The PM phase is used at first in order to check whether the requested URL is listed in the blacklist of the PhishTank database. On the other hand, the UA phase is used for investigation and checking for the absence of undesirable symbols in uniform resource locators (URLs) of the requested website. Finally, the DP phase is used as another level of investigation in order to check for the presence of the requested URL in the desirable whitelist. The obtained results show that the proposed framework is deployable and capable of filtering various types of phishing website by maintaining a low rate of false alarms.

## 1. Introduction

An internet scam is a common social engineering attack used to disclose private and sensitive data by misleading users without being found out [1]. The ultimate aim of scamming websites is to obtain confidential details illegally, such as the username, password, and number of accounts [2]. A phishing attack is the common form of internet scams, which can be launched using several techniques, such as fraudulent emails, social chatting platforms, short message service (SMS) and voice/video over internet protocol (VoIP). Users usually have several profiles, such as a bank accounts, emails, and social networking on various websites. The most susceptible targets of this assault are the innocent users who are unaware of phishing attacks and their fraudulent manners.

An Anti-Phishing Working Group Organization statistics report [3] indicates that during the second quarter of 2022, there were 1,097,811 phishing attacks. According to the Microsoft Security Intelligence report for the first half of 2022, “credential phishing schemes are increasingly prevalent and pose a significant threat to all users, as they target all inboxes without discrimination, luring users to fraudulent websites” [3,4]. Cybercriminals use phishing websites for earning money illegally. Therefore, billions of dollars have recently been found looted from the United State, Russia, and Eastern European banks. AlDairi and Tawalbeh in [5], and Ijaz et al. in [6], have highlighted that phishing attacks recently represented serious threats against the safety of the community within smart cities. Using phishing attacks, cybercriminals target users’ emails to capture their credentials with relevance to their accounts that are widely used within smart city’s applications. This way, the cybercriminal can use the credentials gained from phishing assault to illegally access systems of the smart cities for malicious purposes. Once cybercriminals could successfully obtain an illegal access to the smart city’s systems, they are able to manipulate things with high critical danger against the safety of the community, such as nuclear system damage, gas leakage, or modifying train-signalling systems.

In particular, a phishing website attack is executed through several steps, as illustrated in Figure 1. First, the cybercriminal creates a spoofed webpage, completely similar to the target website [7]. The complete similarity between the spoofed and original webpages has a big effect on deceiving the victims. Second, the URL of the spoofed webpage is sent to the victim using one of the social engineering attack techniques [8]. In most cases, the cybercriminal places the URL of the spoofed webpages on popular websites as well. Third, by clicking on the fake URL, the victim’s request will be forwarded to the website of the cybercriminal, rather than to the real site page. Fourth, the victim’s credentials are collected by the phishing website. Finally, the cybercriminal uses the collected credentials to hack the victim’s account through the targeted website.

Several research studies have been carried out recently to identify the phishing website attacks. Such studies, however, cannot avoid the advanced hacking of phishing websites adequately. Moreover, the utilization of different media communication, including social networks, raises the amount of web-based assaults. In [9], Jagatic et al. reported that a social network is used to execute 70% of phishing website assaults. The absence of awareness and training on phishing website attacks assists attackers to practice their attacks successfully. Failure to differentiate between the spoofed and genuine websites is still a challenge in the existing systems of phishing website detection. The existing systems of malware, an anti-phishing software, do not effectively detect the phishing website attacks. Moreover, the digital certificate (DC) and Secure Socket Layer (SSL) are unable to immunize web users against the real-time phishing attacks. Certain types of DCs and SSLs can be fabricated, while the entire webpage content seems to be legitimate [10].

This paper develops a real-time framework for phishing website detection (ScamFree), mimicking the honeybee’s defense nature mechanism. ScamFree is basically an enhancement for the PhishSys system, which we have previously proposed in [11] for the detection of phishing websites. The focus of ScamFree is to have a smart city, free of phishing attacks. The ScamFree framework is designed to filter out phishing websites using three primary investigative phases, namely PhishTank-Match (PM), Undesirable-Absent (UA), and Desirable-Present (DP). The PM technique is used in the first phase, in order to investigate the suspicious website and classify it as a phishing website in case its URL has been previously included in the PhishTank blacklist [12]. In the UA phase, the URL of the requested website is further investigated and filtered as a phishing website if any undesirable symbol presents within its content. The DP technique is used in the third phase to ensure that the suspicious URL, which has not been classified as a phishing website in the previous phases, is not a new phishing website. In the DP phase, the suspicious URLs will be inspected based on the whitelist of website URLs. Figure 2 illustrates the main process of phishing website filtration implemented in the proposed framework, ScamFree.

The rest of this paper is structured as follows: Section 2 presents the related work. The proposed ScamFree framework is explained in Section 3. Section 4 describes the implementation setting and experimental results. Finally, Section 5 concludes the paper and highlights the future work.

## 2. Related Works

This section critically discusses the most related work of phishing website attacks. Several studies on phishing website attack have been conducted over the past few years. The main aim of such studies is to achieve the most effective and accurate detection of phishing website attacks. The existing studies that used in phishing website attack prevention are mainly categorized into three classes: heuristic-based, blacklist-based, and machine-learning-based approaches, as illustrated in Figure 3.

### 2.1. Heuristic-Based Approach

Heuristic-based approaches elaborate on the HTML or URL signature in identifying the phishing websites. Many researchers have conducted studies on phishing website detection based on this type of approach. For instance, in [13], Chou et al. have introduced a heuristics-based technique, named SpoofGurad, to be used as an anti-phishing browser plug-in. A stateless page evaluation has been implemented in this technique for state full-page evaluation. Moreover, this technique computes the spoof index value as a way to examine the outgoing post data. The authors have identified a threshold value that helps classify the webpages into phishing/normal pages. When the calculated spoof index is more than the identified threshold value, the user will be notified about this page. On the other hand, when the spoof index value is less than the pre-defined threshold value, the webpage will be considered a genuine page. The SpoofGuard technique, however, is still flawed, as it may generate a large number of false positive alarms during the phishing attack.

Studies in [1,14] analyze the URLs of the suspicious webpages to discriminate among genuine and phishing websites. These studies rely on distinguishing the features of the individualities that could be used in perceiving the phishing web pages. Utilizing these types of website phishing detection techniques, spoofing assault could be recognized and reported when it is launched. Thus, it will assist in degrading the need to sustain a blacklist that could inquire time in identifying and produces more complexity. Nevertheless, these techniques are also producing high false negative rate due to the number of malicious webpages which are categorized as legitimate. 

The similarity-based index among the original website and the spoofed ones could be identified using content-based approaches. This type of approach calculates the similarity between websites that have overlapping contents. Generally speaking, we can achieve an adequate level of accuracy with low false alarms in identifying the phishing websites by using the content-based approach. In [15], Zhang et al. have proposed the CANTINA approach, as it works based on content similarity in identifying phishing websites. In particular, CANTINA uses term frequency/inverse document frequency (TF-IDF) for detecting phishing websites. Utilizing the CANTINA approach, the false positive rate was successfully reduced. That was due to the use of the TF-IDF method in retrieving data and performing text mining. The detection level of the CANTINA approach has shown its efficiency in detecting around 97% phishing websites, with 6% false positive alarms. This study has also reported that with merging a heuristics approach to TF-IDF, it could manage to catch around 89% of malicious websites while reducing the rate of false positive alarms to 1% only. Although the CANTINA approach can effectively recognize a phishing website, it has some limitations in dealing with the hidden text within the HTML script file. Moreover, the CANTINA approach inquires for wider-scale deployment and evaluation to improve its functionality in phishing website detection. 

Dunlop et al. in [16] have proposed a GoldPhish approach, which is considered under the content-based category. This approach utilizes Google as a search engine in identifying fishing websites. The main assumption of the GoldPhish approach is that phishing websites do not typically last for a long time. This approach works on taking images for the active webpages in the internet browser of users. Afterward, the optical character recognition technique was utilized to transform the collected images into text form. As a way of identifying the possibility of phishing attacks and analyzing the page rank, the GoldPhish approach has made use of the transformed text as an input into the search engine. Based on the performance analysis of the proposed GoldPhish approach, it has been observed that such an approach could demonstrate effectiveness in mitigating the false positive alarms and detecting new phishing websites. In spite of this improvement, the GoldPhish approach has its limitations in terms of time delay in exploring webpages. Alongside that, this approach could be susceptible to attacks on Google’s PageRank algorithm, as well as Google’s search service.

Spoofguard [17], which relies on similarity-based validation, uses a multi-level approach to verify domains. This involves checking the domain against recently accessed domains to detect any subtle modifications that the user may not notice, as well as inspecting the URL for suspicious embedded usernames and invalid port numbers. However, these methods have a drawback, as they are unable to keep up with the continuous creation of new phishing websites.

A recent work was proposed in [18] for the filtration of phishing websites. It focuses on distinguishing between the genuine and phishing websites by checking the uniform resource locators (URLs) of the visited websites. Checking the suspect URLs based on particular features, to verify whether that webpage is a phishing page, performs the inspection process. In case the suspect URLs are detected as a phishing website, it will be reported for prevention. The work investigates the suspect URLs based on a limited set of features, which may be included in the URL content. This will therefore affect the performance in detecting the zero-day phishing websites. Table 1 concludes this subsection by summerizing the pros and cons of heuristic-based approaches.

### 2.2. Blacklist-Based Approach

As another strategy that is widely used nowadays in identifying phishing websites, a blacklist-based approach has been implemented as an anti-phishing technique. Using this approach, the system should obtain an up-to-date blacklist for the common phishing websites. Altogether, entries that are denied access are allocated in such a phishing blacklist [14,18]. Accordingly, users avoid accessing websites that exist in the blacklist. In particular, blacklist-based approaches focus on tracing the URLs of malicious websites as a way to preserve and generate the blacklist. These URLs could be traced out from the spam email, phishing emails in users’ accounts, or from the associations that assist in preventing phishing assaults, such as PhishTank and Anti-Phishing Working Group (APWG). As soon as a URL is stated, it will then get verified before being inserted into the blacklist. 

As a representative of phishing detection techniques based on the blacklist approach, Net Craft Toolbar [19] is discussed to provide a better understanding. This toolbar discovers the website’s security threat using certain criteria, such as timestamps of accessing the website, time of setting the Net Craft web server survey, rating the risk scale, and the name of the organization, as well as the country of the hosting the website. The Net Craft Toolbar is useful to minimize the number of phishing website assaults. Moreover, this toolbar could also assist in preventing the users from auto download malwares that can be utilized by the hackers in gathering personal data of internet users. It is also useful in mitigating annoying attacks, such as DNS poisoning and pop-up windows, which hackers would use to hide the address bar. In spite of the positive points behind the Net Craft Toolbar in mitigating the known phishing assaults, it still has a limitation in detecting the unknown phishing URLs. It requires performing an endless update on its blacklist database in order to include the newly discovered phishing URLs. 

In [20], Rao, R.S. and Pais, A.R proposed an enhanced technique for detecting phishing websites, using discriminative features extracted from their source code. This enhanced blacklist approach aims to identify phishing sites that mimic existing websites with altered content. Each phishing website is assigned a unique fingerprint, generated using the Simhash algorithm based on a set of proposed features. The fingerprint is derived from file names of request URLs (js, img, CSS, favicon), pathnames of request URLs (CSS, scripts, img, anchor links), and attribute values of tags (H1, H2, div, body, form). The method was tested, and the results show that it achieves an accuracy of 84.36% in detecting phishing sites that replicate other phishing websites with manipulated content, while maintaining a zero false positive rate. This technique is similar to traditional blacklists, but with the added advantage of being able to efficiently detect replicated and manipulated phishing sites. The pros and cons of blacklist-based approaches are summarized in Table 2, bringing an end to this subsection.

### 2.3. Machine-Learning-Based Approach

In the literature relatingphishing website detection, there are a numerous number of machine learning techniques that can be directly utilized in the context of phishing URL identification, such as Decision Trees [22], Naïve Bayes [23,24], Logic Regression, and Support Vector Machine [24,25]. The machine-learning-based studies of detecting phishing websites focused on URL detectors. In particular, these techniques use a collection of URLs as training data, and creates a prediction function to classify a URL as phishing or benign. Consequently, these techniques have the capability to generalize brand-new URLs, unlike heuristic- and blacklists-based strategies. 

For machine learning, the main requirement for training a model is training data. The identification of phishing URL is definitely subjected to the numbers of URLs that are used in the training process. Moreover, the precise process of extracting the optimal features for both phishing and genuine URLs to be used as training data will increase the detection accuracy. The more common lexical features used include the statistical properties of the URL string, its length, and the length of each component (host name, primary domain etc.) [25]. PhishAri, a browser extension that utilizes the Random Frorest classification technique to identify phishing URLs within tweets based on URL features has been developed in [26]. PhishAri achieved an accuracy of 92.52%. The authors in [27] reported accuracies of 97.36% using various supervised machine learning algorithms, including Random Frorest, while an accuracy of 96.17% has been reported by [28]. Convolutional Neural Network (CNN) has been employed in [29] and recorded an accuracy of 98.60%. Although the learning-based techniques have a promising detection in detecting the zero-day phishing URL, they still have limitations relevant to the trained pattern. During the training process, the phishing URL may be fed into the trained pattern, which later classifies the phishing URLs as genuine. Moreover, these techniques are unable to provide a complete diagnosis of undesirable URLs [30].

A number of research studies, such as [31,32], have recently demonstrated that powerful strategies, which can be applied to cybersecurity, could be inspired by natural insects’ behavior systems. A feasible and effective strategy for protecting systems against network intrusions was inspired by the social protection system of honeybees [33,34]. In nature, honeybees struggle to live in environment hazardous with various kinds of threats. This hazardous environment impels the bees to gain practical defense strategy to early discover anomaly activities that may threaten their colony [35].

In the honeybees’ colony, there is a tiny entrance guarded by a particular part of the honeybees, called colony guards. The colony guards are responsible for checking incomers on the entrance to the colony and for preventing them from entering the colony if they do not belong to the colony [36]. In [37], Stabentheiner et al. have stated that honeybee guards distinguish the colony’s nestmates from the non-nestmates by implementing two main methods: Undesirable-Absent (UA) and Desirable-Present (DP). Further information about the UA and DP methods can be found in [33,34]. Phishing website detection systems face the same challenge as the one faced by honeybee guards. As honeybee guards face the difficulty of discrimination between the intruders and the legitimate nest-mate, phishing websites detection systems face difficulty of discrimination between normal and phishing websites.

Our previous work, named PhishSys, has been presented in [11] for detecting phishing website based on the honeybee security system. The PhishSys system uses the honeybee intelligent technique in securing the bees’ colony based on the UA and DP strategies. It filters the spoofing webpages via three investigation stages, which are executed by the three main agents named PhishTank-Match (PM), Undesirable-Absent (UA), and Desirable-Present (DP). Although helpful, ScamFree is presented in this paper to enhance the PhishSys in terms of detection accuracy. Moreover, ScamFree is basically adapted to be utilized for smart cities. Table 3 summarizes the pros and cons of machine-learning-based methods, concluding this subsection.

## 3. Proposed Model Phisfilter

Phishing website attacks happen when the victim is directed to the spoofed website using forged URLs. This section describes the agents and algorithm of ScamFree as a proposed framework for phishing attack detection. In particular, ScamFree consists of three main agents: PM agent, UA agent, and DP agent. Further details about ScamFree agents are provided in the following subsections.

### 3.1. PM Agent

This agent is used to investigate the suspicious URL by checking if it exists in the blacklist of the PhishTank database. PhishTank is one of the public databases of phishing URLs [12]. The PhishTank API accepts an HTTP POST request, along with the query URL, and returns a JavaScript Object Notation (JSON) object in response, which tells whether the query URL is phishing or not. The use of the JavaScript code is to check the suspicious URL links, the results showing that the URL is either a secure page or suspicious. The function of checking a URL is to send a POST request to PhishTank in order to use their API. The function will then process the returned response and determine whether the page is a verified phishing website or not, by looking at the response result. The function would then return true if the URL is verified as belonging to the PhishTank blacklist, or false if not.

### 3.2. UA Agent

This agent is responsible for further investigating the content of the URL, which was filtered by a PM agent in the previous phase. Its main aim is to check if the URL contains one of the undesirable features. Ahmed and Sadiq [11] demonstrated that undesirable features are useful in distinguishing the phishing webpages from legitimate ones. These undesirable features are checked with a variety of parameters, such as URL length, IP address, the addition of a prefix or a suffix, the “//” mark for redirecting, and URLs with the “@” sign. Such features are checked by a series of rules to differentiate URLs for phishing websites from URLs to valid sites.

Several phishing site URLs are inserted in front of the actual URLs. An instance of this is http://www.legitimate.com//http://www.phishing.com (accessed on 20 February 2018). This feature checks the position of the “//” symbol in the URL. It would mean that the “//” symbol will appear in the sixth place if the URL begins with “HTTP.” However, if the URL requires “HTTPS,” the mark “//” is in seventh place.

### 3.3. DP Agent

Given the features of the zero-day attack usually not included in the signature databases, a potent defense system should use an alternative strategy to detect the new attacks. The ScamFree framework provides an agent for further investigating a suspicious URL, which does not exist in the blacklist of the PhishTank database, and verifying if that URL is a new phishing website. Checking the whitelist of legitimate URLs, which is the opposite of a blacklist and contains a list of known trusted sites, does this. More details about the anti-phishing methods using a whitelist database are described in [39]. ScamFree uses the mentioned agents in order to achieve an integrated process for filtering the phishing websites. Figure 4 illustrates the main methodology of the ScamFree framework.

## 4. Implementation and Results

The ScamFree framework is implemented and tested in Google Chrome explorer. Chrome extension allows for adding functionality in Google Chrome without diving deeply into native code. Nowadays, Chrome provides extensions with many purpose APIs for developers to enhance the user browsing experience. This method requires the Chrome platform APIs to perform their function. Those platforms applied in this method are JavaScript APIs Manifest File format and Permission warnings. A JSON-formatted manifest file, named manifest.json, was included as part of the Chrome extension package. 

The use of JavaScript code is to check the URL link, it either being for a secure page or a genuinely suspicious URL site. The function of checkURL is to send a POST request to PhishTank in order to use their API. The function will then process the returned response and determine whether the page is a verified phishing website or not. The function would then return true if the URL is verified as belonging to a phishing website, and false if not. The PhishTank database is one of the familiar public crowd-sourced databases of phishing URLs, containing blacklist data collection [12]. Upon submitting an HTTP POST request to the PhishTank API, PhishTank returns a JSON object that identifies whether or not the query URL is phishing.

The outcome of this method is displaying pop up alert as the result to notify the user that the targeted website is either a secure page or a suspicious URL site. When the page is safe, Chrome will notify “safe” page only in the console and redirect to the targeted page, which allows the user to continue browsing. If the URL website is suspicious, Chrome will notify the user with pop up alert with message “This is a suspected phishing page!!!” This to make the user aware that the targeted page is a suspicious website that is probably a phishing site, luring the user to give their personal information. For detecting URL phishing websites by using Chrome extensions, the extension is loaded into the user’s browser, as illustrated in Figure 5.

The proposed framework, ScamFree, is examined to validate its ability in identifying the spoofing website. To this end, a set of 200 URLs is used, in which 150 are phishing websites and 50 are genuine webpages. The utilized URLs are picked up randomly by PhishTank [12] and Google Safe Browsing. Table 4 shows some selected samples of phishing URLs taken from [11] to be used in this experiment. For every URL, ScamFree inspects its content to verify if it is matched with the known features of the phishing webpages. 

The obtained result demonstrates that ScamFree classifies 49 of the URLs as genuine websites and the other 151 URLs as phishing websites. The obtained result is illustrated in Figure 6.

The performance of ScamFree is evaluated by measuring the accuracy of phishing attack detection and the percentage of false positive alarms. The detection accuracy is measured using Formula (1) as described in [40].
(1)$$\mathrm{Accuracy}=\left(\frac{ɛ+έ}{ɛ+έ+ἓ+ὲ}\right)\times 100\%,$$
where ɛ is true positive, έ is true negative, ἓ is false positive, and ὲ is false negative. More details about ɛ, έ, ἓ, and ὲ measurements are found in [41]. The false positive alarm is measured by calculating the percentage of genuine websites, which were falsely filtered as phishing websites, with respect to the percentage of all genuine websites. The false positive alarm percentage is calculated using Formula (2) as described in [40].
(2)$$\mathrm{False}\;\mathrm{Positive}\;\left(ἓ\right)=\left(\frac{ἓ}{ἓ+έ}\right)\times 100\%$$

The results shown in Figure 6 demonstrates that the ScamFree framework detects the phishing website with an accuracy of 0.995, meanwhile the percentage of false positive rates does not exceed 0.007.

The proposed framework is evaluated in this section by comparison with existing methods, which have been previously discussed in the related work section. The criteria listed in Table 5 are utilized to compare the performance rate of the proposed framework. Table 6 shows that the honeybee-based framework proposed in this paper overall has a competitive result. In the table, the symbol “*” indicates high results, “+” indicates medium results, and “−” indicates low results. Results of the state-of-the-art reported in this table are obtained from their respective articles.

## 5. Conclusions and Future Work

ScamFree detects the phishing websites through three cascading filtration phases, which are performed by the three main agents: PhishTank-Match (PM), Undesirable-Absent (UA), and Desirable-Present (DP). These agents inspect the suspicious URLs using a set of features to ensure that the visited website is genuine. In particular, the suspicious URLs are inspected to discover if they are included in the PhishTank blacklist, any undesirable symbol encompassed within their content, or they are matched with the whitelist of genuine websites. The advantage of ScamFree is its ability to mitigate the limitations of the existing approaches. ScamFree could actively mitigate the high false rate by using several phases of investigation. On the other hand, ScamFree avoids the high overhead against system performance by considering the filtration strategy. In ScamFree, there is no need to investigate all requested URLs, instead, only the suspicious ones that will be filtered and investigated accordingly. Furthermore, investigation of the suspicious URL against the whitelist websites is only required for short-listed URLs to detect the zero-day phishing websites. The findings of this paper have demonstrated our proposed framework’s ability to identify the fake webpages based on their URLs.

For future work, we are planning to improve the method of investigating the URL content, which is through the UA agent. Investigation of the undesirable features will be improved using an intelligent method, which will train the framework to detect the undesirable features based on desirable and undesirable patterns.

## Figures and Tables

**Figure 1 sensors-23-04284-f001:**
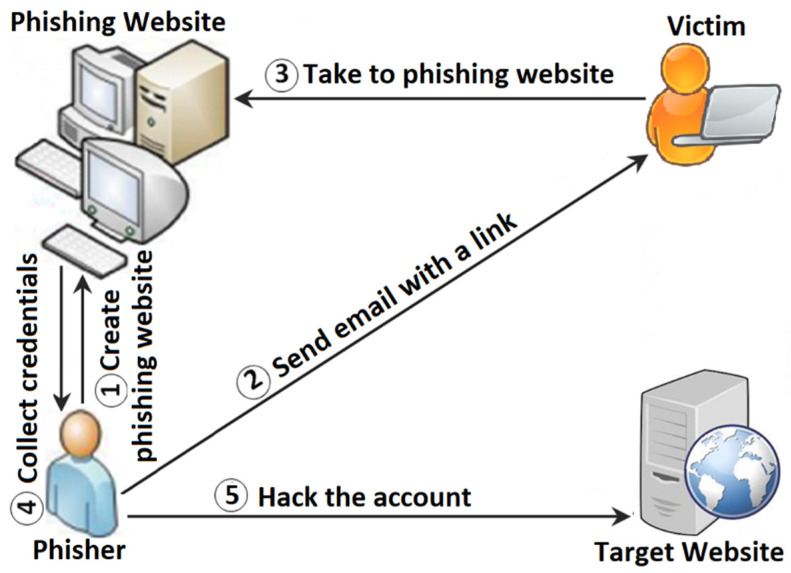
Steps involved in a phishing website attack.

**Figure 2 sensors-23-04284-f002:**
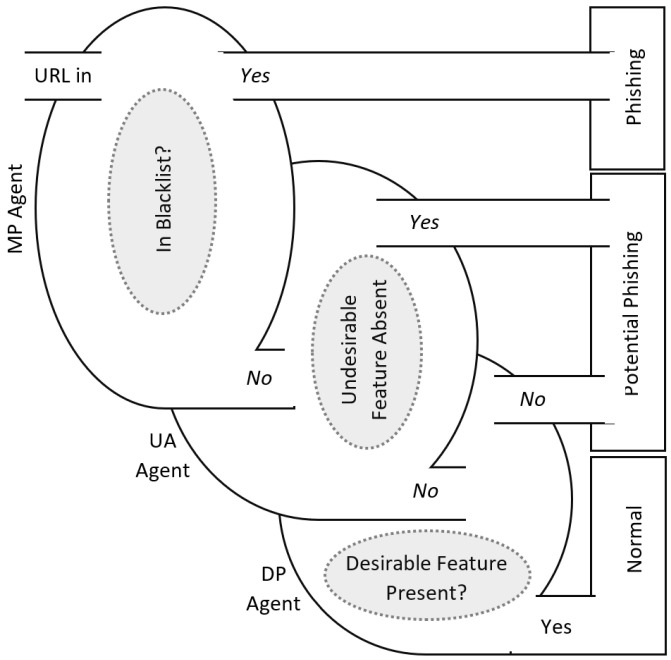
ScamFree filtration process.

**Figure 3 sensors-23-04284-f003:**
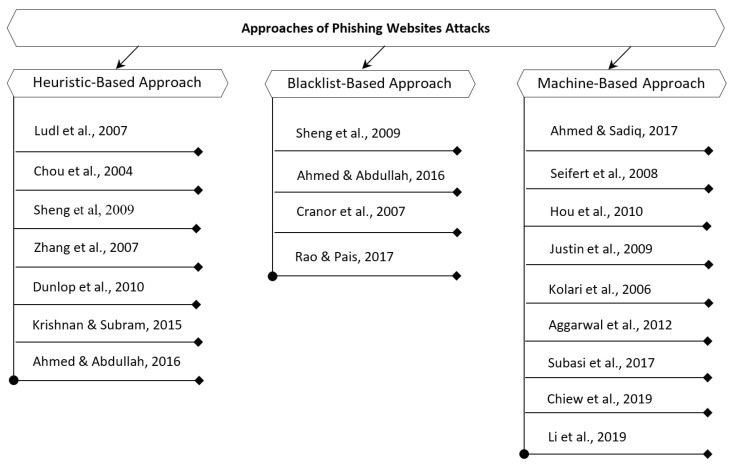
Approaches of phishing website detection [1,11,13,14,15,16,17,18,19,20,21,22,23,24,25,26,27,28,29].

**Figure 4 sensors-23-04284-f004:**
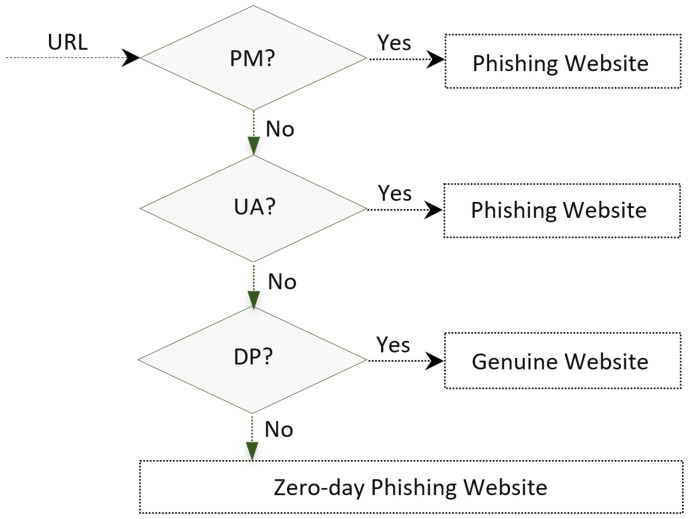
ScamFree algorithm.

**Figure 5 sensors-23-04284-f005:**
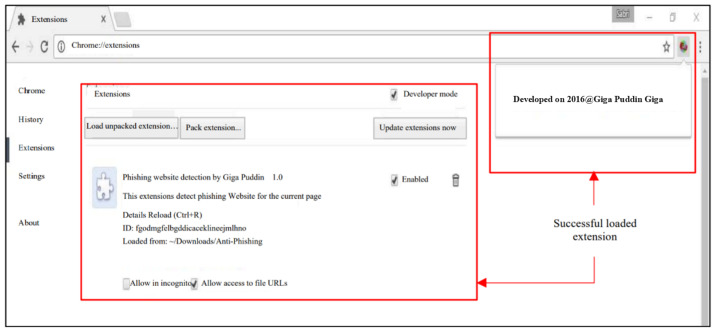
ScamFree extension in Google Chrome [11].

**Figure 6 sensors-23-04284-f006:**
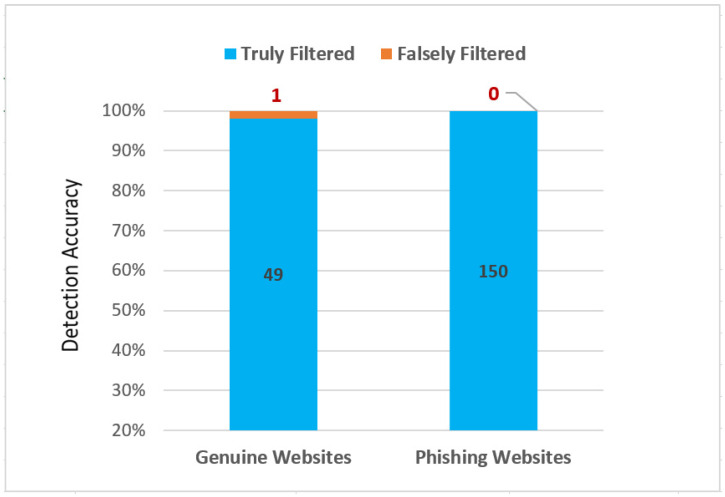
Phishing detection accuracy.

**Table 1 sensors-23-04284-t001:** Pros and cons of heuristic-based methods.

Pros	Cons
Can be more effective at detecting phishing websites that may not have been added to blacklists or other databases.	May generate false positives, flagging legitimate websites as phishing sites and causing inconvenience and frustration for users.
Can analyze various features of the website, such as the domain name, URL structure, and content, to identify phishing patterns and indicators.	May not be able to detect some sophisticated phishing attacks that employ advanced techniques to evade detection.
It is somewhat flexible and responsive to new threats and emerging trends related to phishing attacks.	May require significant computational resources and can be slower than other detection methods due to the delay in exploring webpages.
	URL-based methods have limited information and cannot entirely depict the attributes of the illicit websites.

**Table 2 sensors-23-04284-t002:** Pros and cons of blacklist-based methods.

Pros	Cons
Effective at blocking known phishing websites that have been added to the blacklist.	Less effective in detecting new or unknown phishing websites that have not yet been added to the blacklist.
Can be updated frequently to add new phishing websites to the blacklist.	It can generate false negatives, failing to detect some phishing websites that are not on the blacklist.
Easy to implement and do not require significant computational resources.	May result in over-blocking [21], flagging legitimate websites as phishing sites and causing inconvenience and frustration for users who are prevented from accessing legitimate websites. This can undermine the effectiveness of the detection method if users start to ignore warnings or disable the detection altogether.

**Table 3 sensors-23-04284-t003:** Pros and cons of machine-learning-based methods.

Pros	Cons
Capable of detecting phishing websites with an accuracy rate that exceeds 98.3% [29,38].	Overfitting can occur when machine learning algorithms are trained on a limited set of data, leading to poor performance on predicting future observations.
Machine-learning-based methods can quickly analyze large amounts of data to detect characteristics of phishing websites.	Some machine learning algorithms are difficult to interpret, making it challenging to understand why a certain decision was made.
Can adapt to changing environments and update their algorithms to detect new threats. This can help to reduce the number of false positives and false negatives over time.	Machine learning models rely on historical data to identify patterns and anomalies, which may lead to less effectiveness in detecting new phishing attacks.
Can be trained on large datasets, making them scalable and able to check the visited website against a large number of website requests.	Datasets should be large enough for the system to train on, and they should represent the types of phishing attacks it may encounter. Obtaining and labelling these datasets can be time-consuming and resource-intensive.
	Attackers can intentionally try to manipulate machine learning models by feeding them malicious data that has been designed to evade detection.

**Table 4 sensors-23-04284-t004:** Samples of normal and phishing URLs [11].

Normal Sites	Fake Websites/Phishing Websites	Accessed on
Facebook.com	http://appssecure.at.au/facebook.html	(20 February 2018)
Google drive	http://scredlble.com/login/GDrive/1ff71c3539f0ba9ab99eeb75fe36c5a2/	(20 February 2018)
Paypal.com	http://support-paypai.com.itunesverificationhelp.ga/signin/webapps/6282b/websrc	(20 February 2018)
Adobe.com	http://onlinksoft.org/dragon1/products/adobe.php?email=abuse@the-fat-slags.co.uk	(20 February 2018)
Alibaba.com	http://www.footballnewsheadlines.co.uk/wp-admin/css/alibaba/index.php?email=abuse@gmai.com	(20 February 2018)
bankofamerica.com	http://integral.rs/log/verify.html	(20 February 2018)
Netflix.com	http://ebayproduct.com/fonts/UpdateService/netbi/netfIixo/Login/payment.php	(20 February 2018)
Outlook.com	http://access0000.wapka.mobi/index.xhtml	(20 February 2018)
Google Doc	www.lighthousebd.info/BG/index.php	(20 February 2018)
Dropbox.com	http://dropx.allon4dallas.com/71cbb335c02b4e4c65c7cb74bef95278/	(20 February 2018)
Amazon.com	http://triofloridashow.com/ap/amzon/amzon/2baf777d7c13d97bbeb7f3fbcdafb07c/index/web/login.php?action=billing_login=true	(20 February 2018)

**Table 5 sensors-23-04284-t005:** Evaluation criteria description.

Criteria	Description
Accuracy	A measure of the method’s ability to discern legitimate websites from phishing websites.
False Positive Rate (ἓ)	A percentage of legitimate websites that are incorrectly classified as phishing websites. A low FPR is desirable to prevent legitimate websites from being blocked.
False Negative Rate (ὲ)	A percentage of phishing websites that are not detected by the system. A low FNR is desirable to minimize the risk of successful phishing attacks.
Speed	A measure of how quickly incoming website requests can be analyzed and classified.
Scalability	A method’s ability to handle the growing number of website requests; it is important for web browsers that experience a rapid growth in user traffic.
Robustness	Refers to the methods’ ability to continue functioning under different types of attacks.
Usability	Refers to how easy it is for users to interact with the system.

**Table 6 sensors-23-04284-t006:** Evaluation criteria.

Comparison Metrics	Accuracy	False Positive	False Negative	Speed	Scalability	Robustness	Usability
[1]	+	−	−	+	+	−	*
[13]	+	−	−	+	+	−	*
[14]	+	−	−	+	+	−	*
[15]	+	−	−	+	+	−	*
[16]	+	−	−	+	+	−	*
[17]	+	−	−	+	+	−	*
[18]	+	−	−	+	+	−	*
[19]	+	−	−	+	+	−	*
[20]	+	−	−	+	+	−	*
[11]	*	+	*	+	+	*	−
[22]	*	+	*	+	+	*	−
[23]	*	+	*	+	+	*	−
[24]	*	+	*	+	+	*	−
[25]	*	+	*	+	+	*	−
[26]	*	+	*	+	+	*	−
[27]	*	+	*	+	+	*	−
[28]	*	+	*	+	+	*	−
[29]	*	+	*	+	+	*	−
Honeybee framework	*	+	*	+	+	*	−

“*” sign indicates high results, “+” sign indicates medium results, and “−” sign indicates low results.

## Data Availability

The data we used is public and available online.

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
