# Peer review of "A Honeybee-Inspired Framework for a Smart City Free of Internet Scams"

_sensors, 2023, doi:10.3390/s23094284_

Round 1

Reviewer 1 Report

1. Summarize abstract 

2. Explain figure 1 in detail 

3. Make your own version of figure 2 

4. Related work section is too much short try to add some limitation table in this section 

5. Results are not properly discussed 

6. More simulation graphs and figures need to be added 

7. Add separate section of future directions 

8. Formula 1 and 2 are poorly discussed 

9. Conclusion need to be summarized

Author Response

1. Summarize abstract >> Done as requested.

2. Explain figure 1 in detail>> It is explained in Page 2.

3. Make your own version of figure 2 >> Done as requested.

4. Related work section is too much short try to add some limitation table in this section >> We have made a major revision in the realted work section to enlarge its content by adding more relevant studies as well as adding tables summerizing the pros and cons of the analyzed studies.

5. Results are not properly discussed >> Results has been updated and improved by adding more content to describing the evaluation process.

6. More simulation graphs and figures need to be added >> We have added 2 tables.

7. Add separate section of future directions >> The future work is added to the conclusion section as these 2 sections have small sizes.

8. Formula 1 and 2 are poorly discussed >> have been reformatted.

9. Conclusion need to be summarized >> The conclusion is summmerized as requested

Reviewer 2 Report

1.     What is the novelty of the paper?

2.     More factual and latest references need to be added.

3.     All Figures and text in some places need to be justified.

4.     A separate section for abbreviations should be added.

5.     Check the paper thoroughly for typo/grammatical errors.

6.     Discussion of the results need to be clearer and more elaborated and should be done in a different section apart from conclusion.

7.     The abstract of the paper need to be a little crisp and concise.

8.     Proper spacing after the sub-headings and in Figure captions specifically in Figure 5, should be done.

9.     The paper is not formatted properly.

10.  All text should be in the same format/font, like ‘ScamFree’ is in Italic in some places.

Author Response

  1. What is the novelty of the paper? >> Is the utilization of honeybee - based method for detecting internet scams in smart city.
  2. More factual and latest references need to be added. >> References are updated with recent ones from 2022 and 2023.
  3. All Figures and text in some places need to be justified. >> justified as requested.
  4. A separate section for abbreviations should be added. >> done where possible.
  5. Check the paper thoroughly for typo/grammatical errors. >> done.
  6. Discussion of the results need to be clearer and more elaborated and should be done in a different section apart from conclusion. >> results has been revised and updated.
  7. The abstract of the paper need to be a little crisp and concise. >> Abstract is revised accordingly.
  8. Proper spacing after the sub-headings and in Figure captions specifically in Figure 5, should be done. >> all are fixed.
  9. The paper is not formatted properly. >> Formatting is fixed.
  10. All text should be in the same format/font, like ‘ScamFree’ is in Italic in some places. >> Addressed.

Reviewer 3 Report

In this paper, the authors proposed a honey bee framework for a smart city. This research topic is very interesting. 

1) The introduction is very long. Therefore it should be shortened. The uncessary-related work should be shifted to section 2. In addition, the motivation, contribution, and benefits of this research should be mentioned in the introduction section.

2) In related work it would be great if you mention these studies along with the pros and cons in figure 3.

3)The figure quality should be improved in the final version.

4)The simulation tool details should be mentioned in section 4.

5) The mathematical equation was written in the form of images. Please correct it according to the journal template and should be corrected in the final version.

6) future work should be mentioned in the conclusion section.    

7) The references are not enough, please cite the most recent studies for example;

Author Response

1) The introduction is very long. Therefore it should be shortened. The uncessary-related work should be shifted to section 2. In addition, the motivation, contribution, and benefits of this research should be mentioned in the introduction section. >> Introduction is revised accordingly.

2) In related work it would be great if you mention these studies along with the pros and cons in figure 3. >> This comment has been addressed by adding additional studies to the related work sections and also adding tables decribing the pros and cons of the differnt methods.

3)The figure quality should be improved in the final version.>> This figures are improved.

4)The simulation tool details should be mentioned in section 4. >> This was described in page 10.

5) The mathematical equation was written in the form of images. Please correct it according to the journal template and should be corrected in the final version. >> We have rewritten the equations to address this comment.

6) future work should be mentioned in the conclusion section.   >> This addressed as well. 

7) The references are not enough, please cite the most recent studies for example; >> We have addressed this comment by adding more recent relevant references. We have increased the reference about 25%.

Round 2

Reviewer 1 Report

All comments are incorporated and paper must be processed further

Reviewer 2 Report

Manuscript can be accepted for the publication.

Reviewer 3 Report

The authors has incorporated all necessary changes.